

# Evaluation of sesame (*Sesamum indicum* L.) varieties for drought tolerance using agromorphological traits and drought tolerance indices

Getahun Yemata[1] and Tewachew Bekele[2]

[1] College of Science, Department of Biology, Bahir Dar University, Bahir Dar, Ethiopia
[2] Biology, North Achefer District, Liben Senior Secondary and Preparatory School, Liben, Amhara National Regional State, Ethiopia

## ABSTRACT

Sesame (*Sesamum indicum* L.) is an important cash crop cultivated under rain-fed conditions where it contributes a significant proportion of Ethiopia's foreign exchange earnings. However, its productivity is constrained by drought stress. The present study aimed to evaluate the agromorphological and yield performance of sesame varieties and to identify drought tolerant varieties using drought tolerance indices. The sesame varieties were evaluated under well-watered (WW) and water-stressed (WS) field conditions with a factorial design laid down in randomized complete block design in three replications. The results revealed the presence of a significant variation in agromorphological traits and drought tolerance indices due to water levels, varieties and their interactive effect. On average, a 21.8, 49.6, 48.4, 47.9 and 21.7% reduction was recorded in plant height, number of leaves per plant, leaf length, leaf width and relative growth rate (RGR), respectively under WS condition. Similarly, a significant reduction was found in shoot biomass, root biomass, biological yield, number of pods per plant and seed yield under WS condition. These traits showed an average reduction of 52.2, 72.5, 54.0, 51.9 and 52.8%, respectively compared to WW condition. The highest yield reduction was recorded from wollega under WS condition, while the lowest was from abasena. Wollega variety produced the highest seed yield (kg/ha) under WW condition, while gondar-1 and humera-1 had the highest yield in kg/ha under WS condition. Under both water levels, abasena produced the lowest yield (kg/ha). Moreover, gondar-1 and humera-1 varieties had a comparatively higher values of stress tolerance index (STI), yield stress score index (YSSI), yield potential score index (YPSI), geometric mean productivity (GMP) and mean productivity (MP) that are significantly and positively correlated with yield under WS, indicating higher yield performance under water stress. The biplot analysis clustered the varieties as low yielding (abasena) and relatively above average performing varieties (humera-1, gondar-1 and wollega). According to the rank sum of all indices, humera-1 was identified as drought tolerant, while abasena as the most susceptible and low yielding varieties. Thus, humera-1 followed by gondar-1 were found to be drought tolerant and high yielding varieties. However, further studies focusing on drought tolerance mechanisms of the varieties are recommended.

Corresponding author
Getahun Yemata,
gyemata12@gmail.com

## INTRODUCTION

Sesame (*Sesamum indicum* L.) is an annual oil seed crop widely grown in arid and semiarid tropical and sub-tropical regions (*Baghery et al., 2022*). It belongs to the *Pedaliaceae* family and *Sesamum* genus. The genus consists of 20 species native to Africa and Asia (*Bedigian, 2015*). However, *S. indicum* has been recognized as a cultivated species. It is one of the oldest and most traditional oilseed crops, valued for its high-quality seed composed of 44–57% oil, 18–25% protein, 13–14% carbohydrates (*Borchani et al., 2010*). Evidence about the origin of sesame has been debatable. Nevertheless, most researchers claim that sesame was first cultivated in Africa and later taken to India (*Alegbejo et al., 2003*).

In Ethiopia, the production of sesame is rain-fed, characterized by intensive labour and low levels of inputs (*Coates et al., 2011*). It grows in a wide variety of soil types. However, the crop thrives best on well drained and medium textured fertile soil with pH range of 5 to 8. It also needs adequate moisture for germination and early growth. Precipitation of 300–800 mm per season is necessary for reasonable yields (*Terefe et al., 2012*). Sesame is temperature sensitive that requires hot conditions during growth to produce maximum yield. Photoperiodically, late maturing sesame varieties are short day plants, while the early maturing ones can initiate flower and fruit under both short and long day conditions, but did the best under long day condition (*Nafe et al., 2010*). It shows optimum development and yield at 25 to 37 °C temperature throughout its growth period (*Terefe et al., 2012*). Generally, the crop grows to a height of 1.5 to 2.0 m depending on the variety and growing conditions (*Terefe et al., 2012*).

Sesame is produced in different parts of Ethiopia starting from an elevation of 1,500 meter above sea level. The major producers that contribute over 83% to the national production (*CSA, 2011*), are located in the regions of Tigray (West Tigray), Amhara (North Gondar) and most recently, in Benishangul-Gumuz Region (Metekel). In the years 2005–2012, on average 37% of the country's total seed production is contributed by the Amhara Regional State, 30% from Tigray and 16% from Oromia (*CSA, 2013*). The production of sesame has showed an increasing trend for many years. In the 2014/15 cropping season, 464,000 metric tons of sesame seed was produced. This increased to 487,000 metric tons in 2015/16. It showed 5% increment. The increment has been brought mainly by the expansion of production area (*Francom, 2016*) due to the fast growing nature of the oilseed sector in the country. It has become the second largest source of foreign exchange earnings after coffee (*FAO, 2012*).

The average yield of sesame is low in Ethiopia. For the years 2005–2012, the highest average productivity of sesame for Tigray was about 9 quintals/ha, followed by Amhara region about 8 quintals/ha. This is lower than half of the potential yield of the crop estimated by FAO, which is 16 quintals/ha (*FAO, 2015*). According to *Gelalcha (2009)*, the low sesame productivity was attributed to a combination of various factors. The major constraints

include lack of improved seeds (*Teklu et al., 2021*), drought stress, salt stress, low fertilizer input, biotic stress, heat, indeterminate flowering nature and shattering of capsules at maturity and paucity of knowledge on postharvest crop management practices (*Endale, 2017*). As a rain-fed crop commonly cultivated in arid and semiarid tropics, sesame has been frequently exposed to terminal drought (*Pandey et al., 2021*) and such exposure has reduced grain yield by 52% compared to the non-stressed ones (*Kim, Park & Jenks, 2007*; *Golestani & Pakniyat, 2015*). Moreover, the rainfall distribution in the study area has been significantly erratic that has impeded the productivity of sesame. The crop is sensitive to drought, especially at the vegetative stage (*Boureima et al., 2011*). This is reflected in the changes that occur subsequently in plant metabolism, growth, development and yield. However, the effect of drought is more severe on seed yield than other morphological characters. According to *Kim, Park & Jenks (2007)* sesame yield reduction owes to the decreased number of seeds under drought stress. Research reports have revealed that various sesame varieties show variable responses to drought with some varieties being highly tolerant and others more susceptible (*Boureima et al., 2011*).

Despite the presence of high genetic diversity of sesame in the country, more than 870 accessions (*Teshome, Tesfaye & Bekele, 2015*), studies focusing on evaluation of sesame genotypes under drought stress conditions using agromorphological traits and drought tolerance indices have been scarce. Selecting genotypes with optimal performance under both stress and non-stress conditions from other groups is the best tool for identifying genotypes for drought tolerance (*Pandey et al., 2021*). Several such criteria have been proposed, most of which were used in this study to select the best sesame varieties for drought tolerance and thereby recommend suitable sesame varieties in drought prone areas of the country. We hypothesized that the responses of sesame varieties significantly vary under different levels of water and the best performing variety can be chosen using agromorphological traits and drought tolerance indices. Therefore, the present work was aimed at evaluating the agromorphological and yield performance of sesame varieties under well-watered and water-stressed conditions and identifying the best drought tolerant variety using drought tolerance indices.

## MATERIAL AND METHODS

### Description of the experimental area

The experiment was conducted at Liben senior secondary and preparatory school located in Amhara Region, West Gojjam zone, North Achefer district (Fig. 1). The school is located at 11°41′51″ north latitude and 36°56′35′ east longitude. The area has arid and semi-arid climatic conditions with a soil type suitable for irrigation. The altitude of the district ranges from 1,500 to 1,800 meter above sea level (m.a.s.l). It is also characterized by unimodal rainfall with an average annual rainfall ranging from 1,000 to 1,500 mm. The minimum and maximum daily temperatures are 25 °C and 30 °C, respectively (NADoA, 2013 cited in *Demeke, Mekuriaw & Asmare, 2017*).

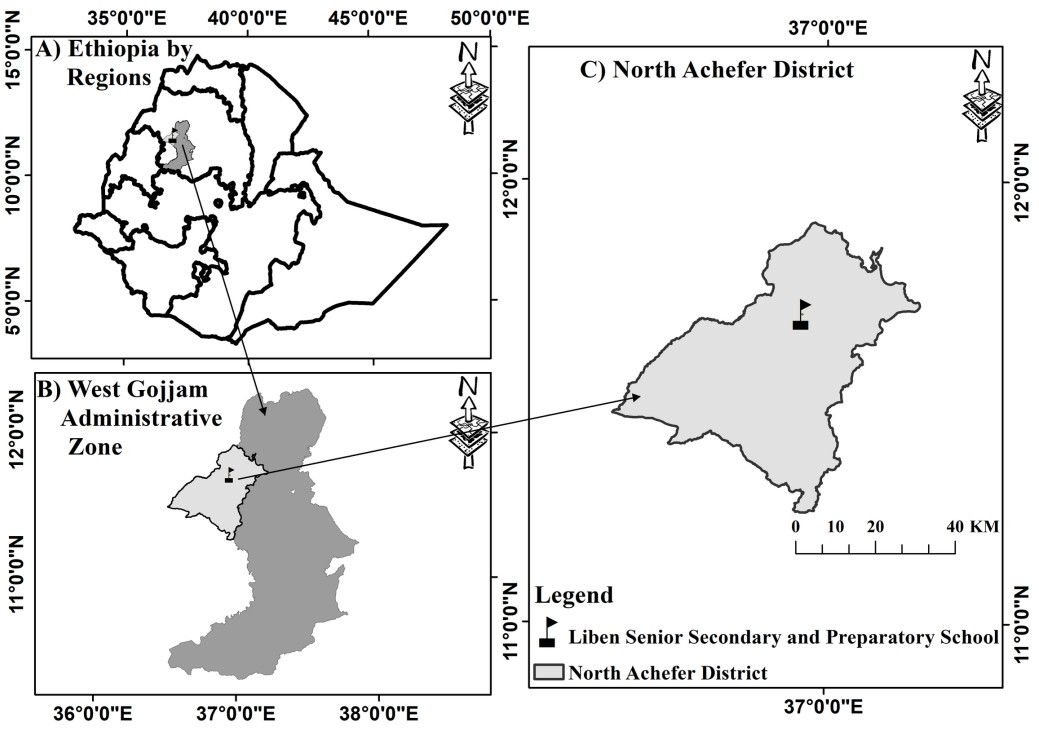

**Figure 1** Map of the study area showing the experimental site.

## Study plant materials

A total of four sesame varieties were used in this study. All of them are released varieties. These are T-85, Kelafo-74, Mehado-80 and abasena. The former had indeterminate form of growth, while the remaining varieties had determinate growth (*Mawcha et al., 2020*). The former two were released in 1976 and the later in 1989. They are the main export varieties and well known by their market names as humera-1, gondar-1, wollega and abasena, respectively. Then here after the market names were used. Humera-1 is characterized by whitish, large and sweet taste seeds, high productivity, high shattering property, 45–50% oil content, maturity time of 110–115 days and adapted to the Humera plains (*Jaleta, 2012*). Gondar-1 has light brown coloration and good uniformity of seed, and a maturity date ranging from 110–120 days. It was released and adapted to the Gode region (*Endale, 2017*). Similarly, wollega is characterized by its small gray seeds, high oil content (49–56%), low sweetness, maturity date of 100–110 days and released and adapted to irrigated areas of Awash valley (*Jaleta, 2012*; *Endale, 2017*). Abasena, on the other hand is characterized by gray, large and sweet taste seeds, high productivity, high shattering property, 44–48% oil content, maturity time of 110–120 days and adapted to high rainfall (*Bekora, 2013*). Seeds of humera-1, gondar-1 and abasena were collected from Amhara Agricultural Research Institute, Gondar Agricultural Center, and seed of wollega variety was collected from local farmers around the study area.

## Experimental design and treatments

The experimental field was tilled using human power. All the weeds and large plant debris were removed manually from the experimental field. Then, it was exposed to sunshine for six days prior to the next tilling. The land was ploughed again and again until smooth soil particles were obtained as this is necessary for the better growth of the crop. The field was ploughed, leveled, ridged and divided into plots before sowing.

The field experiment was carried out using factorial design laid down in a randomized complete block design (RCBD) with three replications. The size of the experimental field was 39 m² (9.75 m × 4 m) consisting of 24 plots each with 1 m² area. The replications were 0.5 m far apart. Each replication consisted of eight plots with 0.25 m spacing to prevent lateral penetration of water. Each plot had three rows with 37 cm spacing. There was 18.5 cm spacing between each sesame plant. Around each plot, 13 cm area was left to avoid edge effect. Four holes were prepared along each row and three seeds were hand sown in each. The sesame varieties were assigned to plots randomly by a lottery method. Thinning to one plant/hole was carried out after complete emergence. N and P inorganic fertilizers were applied at the rate of 100 kg/ha (*Zenawi & Mizan, 2019*) and 46 kg/ha (*Gebremariam, 2015*), respectively. All the P and half of the N fertilizers were applied at the time of sowing and the remaining half N at branching stage. The common management practices such as weeding, pesticide and other parasites monitoring and control measures were carried out as needed.

Treatments consisted of four sesame varieties grown under two water levels. The water levels were determined gravimetrically as described by *Turner (2018)*. Bottom perforated pots filled with dry soil were weighed and then watered for adequate period of time. The saturated soil was left overnight to drain the excess water by the force of gravity and then was weighed. At this condition, the soil water was at its field capacity equivalent to pot capacity. The water level at field capacity was considered as well-watered and 50% of the field capacity was taken as water-stressed treatment. The water level of all treatments was maintained to field capacity throughout the vegetative stage, until the imposition of water stress. Water stress was imposed 30 days after sowing (DAS), which represents flower initiation stage. This is because flowering starts 35–45 days after sowing in sesame and continues for 75 to 85 days for early types and 150 days for some varieties to mature (*Terefe et al., 2012*). Sesame plants grown under well-watered treatment in each plot were irrigated with 333 mL of water for every three days and those in the water-stressed received half of this volume at the same time interval.

## Soil analysis

Soil samples representative of the experimental field were randomly collected from different points and thoroughly mixed together to form a composite soil sample. The sample was dried, ground and passed through a two mm sieve and used for further soil property analysis. The pH was determined in 1:2.5 soil/water ratio using pH meter. The total percent of nitrogen was determined by Kjeldahl digestion method, organic carbon content and organic matter were measured by volumetric method as described by *Roper et al. (2019)*.

The analysis result revealed that the soil had a pH of 6.5, total percent nitrogen (0.084), percent organic carbon content (0.49) and organic matter (0.85).

## Data collection
Growth and physiological data of the study sesame varieties were gathered from three randomly selected and tagged plants in each plot. When the tagged plants were lost due to various reasons, data were collected from other plants in the same plot. Growth data were collected 30 and 60 days after water stress imposition, while physiological data were gathered 30 days after water stress imposition

## Growth parameters
Plant height of tagged plants was measured from the ground level to the tip of the youngest leaf and expressed in centimeter (cm). Number of leaves and branches of the same plants were counted and recorded. The length of leaves was measured from the leaf base to the tip and width at the maximum area of the blade. Relative growth rate (RGR) was assessed based on height and determined according to *Dodig et al. (2021)* as $RGR_H = (\log H_{t2} - \log H_{t1})/t_2 - t_1$ where, $H_{t1}$ is height of the plant measured at $t_1$ and $H_{t2}$ is height of the plant measured at $t_2$, $t_1$ and $t_2$ are times of the first and second height measurements, respectively.

## Physiological parameters
### Relative water content
Relative water content (RWC) was estimated according to *Gonzalez & Gonzalez-Vilar (2003)*. The fully expanded third leaf from the top was collected from tagged plants and weighed producing fresh weight (FW). These leaves were preserved in plastic bags and transported to laboratory. Each leaf was floated on distilled water in a Petri dish for 24 hr. The leaves were then blotted gently with tissue paper and weighed again to get the turgid weight (TW). The leaves were then oven dried at 70 °C for 24 hr and the dry weight (DW) was recorded. Finally, the leaf relative water content was calculated as (FW-DW)/(TW-DW) ×100.

## Yield and biomass parameters
Yield, yield related and biomass data were collected after the complete maturation of the crop. The numbers of capsules/pods per plant in each plot were counted from three tagged plants and recorded during data collection period. All the pods from each plant and plot were collected and dried in an oven at 80 °C to constant weight and threshed plot wise. Grain yield was determined for each plot (1 m$^2$) and reported in terms of kg/ha as described by *Nadeem et al. (2015)*.

After harvesting the pods, each plant was uprooted and parts were separated into shoot (stem and root) and root. The roots were carefully detached from soils and thoroughly washed with tap water. The shoot and root parts were dried in an oven at 80 °C to constant weight and weighed separately. Harvest index was calculated from biological yield (shoot and root dry weight) and grain yield in kg/ha according to *Nadeem et al. (2015)* as harvest index = seed yield/biological yield ×100.

## Drought tolerance indices

Drought tolerance indices are important tools that provide better opportunities to select genotypes with good performance under normal and stress conditions. Several of the tolerance indices were calculated as follow:

Stress tolerance index (STI) = $Ys*Yp/\bar{Y}p^2$ ....................*Ahmed et al. (2020)*

Mean productivity (MP) = Ys+Yp/2 ................*Sun et al. (2023)*

Stress tolerance (TOL) = Yp-Ys ..................*Anwaar et al. (2020)*

Yield index (YI) = Ys/ $\bar{Y}p$ ............*Shahrokhi, Khorasani & Ebrahimi (2020)*

Geometric mean productivity (GMP) = ................*Adhikari et al. (2019)*

Yield stability index (YSI) = Ys/Yp ..............................*Kouighat et al. (2023)*

Stress susceptibility index (SSI) = 1-(Ys/Yp)/1- ($\bar{Y}s/\bar{Y}p$)....*Anwaar et al. (2020)*

% reduction = Yp-Ys/Yp*100 ................................*Choukan et al. (2006)*

Yield potential score index (YPSI) = 0.5 (MP + STI) –0.5 (SSI + TOL) ...(*Malinowska, Donnison & Robson, 2020*)

Stress intensity (SI) = 1- ($\bar{Y}s/\bar{Y}p$) ....................*Zare (2012)*

Yield stress score index (YSSI) = 0.5 (STI + SSI) .....................*Malinowska, Donnison & Robson (2020)*

where, Ys is the yield of each genotype under water-stressed condition; Yp is the yield of each genotype under well-watered condition; $\bar{Y}s$ is the mean yield of each genotype under water-stressed condition and $\bar{Y}p$ is the mean yield each genotype under well-watered condition.

## Ranking of sesame varieties

Different drought tolerance indices discriminate different sesame varieties as drought tolerant. Thus, identifying drought tolerant genotypes based on a single index does not produce clear results. To identify desirable drought tolerant varieties, the mean, standard deviation of ranks, and rank sum of all indices were calculated. For screening drought tolerant varieties, a rank sum (RS) was calculated by using the following formula:

Rank sum = Mean ranks (MR) + standard deviation of ranks (SDR)...*Noorifarjam, Farshadfar & Saeidi (2013)*

## Data analysis

All the data were analyzed using the Statistical Package for Social Sciences (SPSS, version 23). Multiple comparisons of means were carried out with Tukey HSD test to see variations between treatments at 30 and 60 days after drought imposition. Factorial Analysis of Variance (ANOVA) was conducted for each trait under WW and WS conditions and results were considered significant at $p < 0.05$. Furthermore, principal component analysis (PCA) was used to characterize trait variation. PCA and correlation analysis between grain yield and drought tolerance indices were analyzed with R software. The biplot was generated using factor analysis and data processing with R package.

## RESULTS

The sesame varieties showed notable variation among most of the traits between WW and WS conditions. The analysis of variance of the data from the field experiment showed that

the differences between treatments and varieties were statistically significant at $p < 0.05$ (See Table 1, Table S1). In this period, plant height was reduced under WS by 23.2, 27.6, 28.9 and 27.4% in abasena, gondar-1, humera-1 and wollega sesame varieties, respectively. The reduction in RGR was nearly similar to the reduction in plant height for all the varieties as it was derived from plant height. Greater reduction was observed in the number of leaves per plant, leaf length and width under WS condition 60 days after stress imposition. In this regard, the number of leaves per plant was reduced by 48.2, 51.2, 48.3 and 50.8% in abasena, gondar-1, humera-1 and wollega varieties, respectively. The reduction in leaf length and width ranged from 43.9–53.9% and 45.4–51.9%, respectively in the studied varieties (Table 1, Table S1). In general, on average a 21.8, 49.6, 48.4, 47.9 and 21.7% reduction was recorded in plant height, number of leaves per plant, leaf length, leaf width and relative growth rate (RGR), respectively under WS condition. The growth of branches was completely suppressed under WS condition in all sesame varieties. An insignificant difference in leaf RWC was observed between plants under WW and WS conditions in all varieties, indicating that leaf RWC is not an important parameter for screening sesame varieties for drought tolerance.

The results also demonstrated a highly significant difference among varieties under the different water levels, 60 days after water stress imposition at $p < 0.05$. Until 30 days, the effect of WS on some growth parameters was not visible. A significant difference was recorded only in the number of leaves per plant between abasena (6.89) and humera-1 (8.00) varieties under WS condition (Table 1, Table S1). Plants of gondar-1 variety performed the highest in terms of number of leaves per plant, leaf width, and plant height under WS condition at 60 days after water stress imposition. This difference was statistically significant at $p < 0.05$ (Table 1, Table S1). Although insignificant, higher numbers of branches per plant were recorded in wollega variety under WW conditions 60 days after stress imposition (Table 1, Table S1), which might have implication to yield. Plants of gondar-1 had also significantly higher number of leaves per plant than abasena under WS condition. It also showed considerably faster RGR than abasena under WW condition. The average reduction in plant height, number of leaves per plant, leaf length, leaf width and RGR of the study varieties was found to be 21.8, 49.6, 48.4, 47.9 and 21.7%, respectively under WS condition. In general, gondar-1 and abasena sesame varieties performed the best and least, respectively under the present study experimental conditions (WW and WS). Plants of humera-1 and wollega showed moderate growth performance under both water levels (Table 1, Table S1).

Plant height showed positive and significant correlation with RGR in all sesame varieties both under WW and WS conditions at $p < 0.01$. Number of leaves per plant and RWC had higher positive correlation with yield in abasena variety under WW conditions (Table S2). In contrast, almost all the parameters had negative correlation with yield under WS condition. In gondar-1 and humera-1 sesame varieties, almost all growth and physiological parameters had negative correlation with yield both under WW and WS conditions except RWC under WS conditions in humera-1 variety. On the other hand, plant height and RGR had a significantly higher positive correlation with yield in wollega variety under WS condition (Table S2).

**Table 1  Growth and relative water content responses of sesame varieties. Mean values ± SE ($n = 9$).**

| Plant parameters | | 30 days after water stress imposition | | | | 60 days after water stress imposition | | | |
|---|---|---|---|---|---|---|---|---|---|
| | | Sesame varieties | | | | Sesame varieties | | | |
| | Treatments | Abasena | Gondar-1 | Humera-1 | Wollega | Abasena | Gondar-1 | Humera-1 | Wollega |
| Plant height (cm) | WW | 42.17 ± 0.30[aA] | 40.67 ± 0.67[aA] | 41.89 ± 0.45[aA] | 41.83 ± 0.41[aA] | 51.33 ± 0.55[aB] | 56.33 ± 1.83[aA] | 54.67 ± 0.44a[AB] | 53.39 ± 0.60a[AB] |
| | WS | 35.89 ± 0.81[bA] | 34.67 ± 0.73[bA] | 35.61 ± 0.44[bA] | 34.78 ± 0.92[bA] | 39.44 ± 0.29[bA] | 40.78 ± 1.23[bA] | 38.89 ± 0.65[bA] | 38.78 ± 0.32[bA] |
| Number of branches per plant | WW | 0.00 ± 0.00[aA] | 0.67 ± 0.33[aA] | 0.67 ± 0.47[aA] | 1.33 ± 0.75[aA] | 0.00 ± 0.00[aA] | 2.00 ± 0.00[aB] | 2.00 ± 0.00[aB] | 2.44 ± 0.44[aB] |
| | WS | 0.00 ± 0.00[aA] | 0.00 ± 0.00[aA] | 0.00 ± 0.00[aA] | 0.00 ± 0.00[aA] | 0.00 ± 0.00[aA] | 0.00 ± 0.00[bA] | 0.00 ± 0.00[bA] | 0.00 ± 0.00[bA] |
| Number of leaves per plant | WW | 9.56 ± 0.75[aA] | 12.67 ± 1.37[aA] | 9.22 ± 0.57[aA] | 10.33 ± 0.82[aA] | 12.44 ± 0.65[aB] | 17.33 ± 1.41[aA] | 13.33 ± 0.67[aB] | 14.44 ± 1.04a[AB] |
| | WS | 6.89 ± 0.39[aB] | 8.67 ± 0.83b[AB] | 7.44 ± 0.38[aA] | 7.56 ± 0.53a[AB] | 6.44 ± 0.44[bA] | 8.44 ± 1.09[bA] | 6.89 ± 0.59[bA] | 7.11 ± 0.75[bA] |
| Leaf length (cm) | WW | 8.72 ± 0.35[aA] | 9.72 ± 0.19[aA] | 8.00 ± 0.49[aA] | 8.89 ± 0.49[aA] | 11.39 ± 0.73[aA] | 12.89 ± 0.78[aA] | 11.00 ± 0.52[aA] | 11.56 ± 0.48[aA] |
| | WS | 5.11 ± 0.31[bA] | 4.67 ± 0.41[bA] | 4.67 ± 0.29[bA] | 4.33 ± 0.24[bA] | 6.39 ± 0.37[bA] | 5.94 ± 0.13[bA] | 5.89 ± 0.11[bA] | 5.89 ± 0.26[bA] |
| Leaf width (cm) | WW | 4.39 ± 0.14a[AB] | 5.39 ± 0.20[aA] | 4.11 ± 0.42[aB] | 4.67 ± 0.34a[AB] | 6.61 ± 0.22[aA] | 6.56 ± 0.24[aA] | 6.11 ± 0.30[aA] | 6.61 ± 0.26[aA] |
| | WS | 2.78 ± 0.15[bA] | 2.93 ± 0.13[bA] | 2.53 ± 0.17[bA] | 2.82 ± 0.11[bA] | 3.43 ± 0.15[bA] | 3.58 ± 0.15[bA] | 3.17 ± 0.12[bA] | 3.32 ± 0.14[bA] |
| Relative water content (%) | WW | 61.40 ± 4.24[aA] | 61.61 ± 2.64[aA] | 65.31 ± 2.93[aA] | 73.77 ± 9.01[aA] | 46.42 ± 3.79[aA] | 42.85 ± 0.82[aA] | 62.24 ± 14.91[aA] | 58.79 ± 7.67[aA] |
| | WS | 50.42 ± 3.10[aA] | 62.34 ± 2.86[aA] | 58.50 ± 4.80[aA] | 81.14 ± 15.68[aA] | 47.79 ± 24.26[aA] | 51.91 ± 4.05[aA] | 38.49 ± 4.04[aA] | 21.43 ± 14.87[aA] |
| Relative growth rate (cm/day) | WW | 0.86 ± 0.01[aA] | 0.94 ± 0.03[aB] | 0.91 ± 0.01a[AB] | 0.89 ± 0.01a[AB] | | | | |
| | WS | 0.66 ± 0.01[bA] | 0.68 ± 0.02[bA] | 0.65 ± 0.01[bA] | 0.65 ± 0.01[bA] | | | | |

**Notes.**

Mean values in a column followed by different small case letters within the same parameter and mean values in a row followed by different upper case letters are significantly different at $p < 0.05$.

**Table 2  Analysis of variance for main and interaction effects of growth parameters.**

| Source | Dependent variable | df | Mean square | F | Sig. | Partial eta squared |
|---|---|---|---|---|---|---|
| Variety | Plant height | 3 | 33.3 | 4.7 | 0.005 | 0.2 |
|  | Number of branches per plant | 3 | 5.4 | 24.3 | 0.000 | 0.5 |
|  | Number of leaves per plant | 3 | 40.1 | 5.7 | 0.002 | 0.2 |
|  | Leaf length | 3 | 3.0 | 1.5 | 0.242 | 0.1 |
|  | Leaf width | 3 | 0.7 | 1.8 | 0.162 | 0.1 |
|  | RWC | 3 | 121.6 | 0.3 | 0.864 | 0.0 |
|  | RGR | 3 | 0.01 | 4.6 | 0.006 | 0.2 |
| Water levels | Plant height | 1 | 3,762.8 | 526.9 | 0.000 | 0.9 |
|  | Number of branches per plant | 1 | 46.7 | 210.3 | 0.000 | 0.8 |
|  | Number of leaves per plant | 1 | 924.5 | 131.3 | 0.000 | 0.7 |
|  | Leaf length | 1 | 580.9 | 275.6 | 0.000 | 0.8 |
|  | Leaf width | 1 | 172.7 | 450.3 | 0.000 | 0.9 |
|  | RWC | 1 | 297.0 | 0.6 | 0.442 | 0.0 |
|  | RGR | 1 | 1.1 | 520.9 | 0.000 | 0.9 |
| Variety * Water levels | Plant height | 3 | 14.4 | 2.0 | 0.121 | 0.1 |
|  | Number of branches per plant | 3 | 5.4 | 24.3 | 0.000 | 0.5 |
|  | Number of leaves per plant | 3 | 7.3 | 1.0 | 0.382 | 0.1 |
|  | Leaf length | 3 | 3.6 | 1.7 | 0.176 | 0.1 |
|  | Leaf width | 3 | 0.12 | 0.3 | 0.814 | 0.0 |
|  | RWC | 3 | 157.0 | 0.3 | 0.813 | 0.0 |
|  | RGR | 3 | 0.0 | 2.2 | 0.103 | 0.1 |

**Notes.**

RWC, Relative water content; RGR, Relative growth rate.

A significant variation in plant height, number of branches, number of leaves per plant and RGR was recorded due to the separate effect of variety and water levels 60 days after water stress imposition (Table 2, Table S4). According to the analysis of the result, variety had no significant influence on the leaf length and width of the studied varieties. The water levels had greater and significant effect than variety. In this regard, the highest reduction in plant height and RGR under WS was recorded by humera-1 variety followed by gondar-1 and wollega, while the lowest reduction was recorded in abasena variety. The difference in reduction due to WS in plant height, number of leaves per plant, leaf length and width ranged from 23.2–27.4, 48.2–51.2, 43.9–53.9 and 45.4–49.8%, respectively (Table 1, Table S1). The biggest reduction in the number of leaves per plant and leaf length was observed in gondar-1 variety followed by wollega. Similarly, humera-1 and wollega varieties showed the biggest reduction in plant height and leaf width under WS condition (Table 1, Table S1). In the same period, the interactive effect of variety and water levels on growth parameters was insignificant except in the number of branches per plant where the synergistic effect of the two independent variables was greater (Table 2, Table S1).

**Table 3  Mean biomass and yield responses of sesame varieties grown under well-watered (WW) and water-stressed (WS) conditions harvested after the complete maturation of the crop. Mean ± SE ($n = 9$).**

| Sesame varieties | Water levels | Shoot biomass (g) | Root biomass (g) | Biological yield (g) | Number of pods/plant | Seed yield/cm$^2$ (g) | Harvest index (%) |
|---|---|---|---|---|---|---|---|
| Abasena | WW | 3.11 ± 0.15[b] | 0.70 ± 0.04[b] | 3.81 ± 0.15[b] | 6.00 ± 0.33b | 2.14 ± 0.15[c] | 56.65 ± 3.82[b] |
| | WS | 1.68 ± 0.14[b] | 0.39 ± 0.08[b] | 2.07 ± 0.17[d] | 3.11 ± 0.35[c] | 1.42 ± 0.17[c] | 68.40 ± 5.22[b] |
| Gondar-1 | WW | 6.11 ± 0.51[b] | 0.78 ± 0.05[b] | 6.89 ± 0.53[b] | 8.67 ± 0.58[b] | 3.67 ± 0.14[b] | 55.84 ± 4.97[b] |
| | WS | 2.56 ± 0.24[b] | 0.11 ± 0.01[c] | 2.67 ± 0.25[c] | 4.44 ± 0.44[c] | 1.81 ± 0.08[c] | 70.63 ± 4.28[b] |
| Humera-1 | WW | 6.00 ± 0.67[b] | 0.44 ± 0.06[b] | 6.54 ± 0.69[b] | 8.22 ± 0.62[ab] | 4.14 ± 0.36[b] | 68.27 ± 6.73[b] |
| | WS | 2.78 ± 0.28[b] | 0.04 ± 0.01[c] | 2.82 ± 0.28[c] | 3.78 ± 0.52[c] | 1.71 ± 0.15[c] | 65.13 ± 7.89[b] |
| Wollega | WW | 6.56 ± 0.44[b] | 0.45 ± 0.07[b] | 7.11 ± 0.42[b] | 8.22 ± 0.70[ab] | 5.18 ± 0.23[b] | 75.02 ± 5.51[b] |
| | WS | 3.23 ± 0.26[b] | 0.14 ± 0.08[c] | 3.41 ± 0.22[c] | 3.56 ± 0.44[c] | 1.66 ± 0.12[c] | 50.66 ± 5.68[b] |

**Notes.**
Mean values in each column that don't share similar letters are significantly different at $p < 0.05$.

## Yield and its attributes of sesame varieties

The sesame varieties demonstrated a statistically significant variation in biomass production, yield and yield related traits under the two water levels (WW and WS) at $p < 0.05$. A significant reduction in shoot biomass, root biomass, biological yield, the number of pods per plant and seed yield was recorded under WS condition among the sesame varieties. These traits showed an average reduction of 52.2, 72.5, 54.0, 51.9, and 52.8%, respectively compared to WW condition. The highest reduction in yield and yield related traits between WW and WS conditions was found in wollega followed by humera-1, while the lowest was found in abasena followed by gondar-1 (Table 3, Table S3). There was insignificant difference in shoot biomass, harvest index and seed yield between water levels in abasena variety (Table 3, Table S3). Significant differences were also recorded among varieties both under similar and different water levels (Table 3, Table S3). Accordingly, wollega, gondar-1 and humera-1 varieties produced considerably greater shoot biomass and biological yield than abasena under WW condition. The variation in shoot biomass between the varieties under WS condition was insignificant at $p < 0.05$. All varieties under the WS condition had significantly lower root biomass than those in WW condition, implying that enhanced root growth and the consequently increased root/shoot ratio is not a drought tolerance strategy in the sesame varieties in the area under study. In this regard, abasena variety had a relatively greater root biomass than the others under WS condition (Table 3, Table S3). Sesame plants of wollega variety produced a relatively higher biological yield followed by gondar-1 both under WW and WS conditions than others (Table 3, Table S3).

In terms of yield and related traits, gondar-1 variety had greater number of pods per plant followed by humera-1 and wollega varieties under WW and WS conditions (Table 3, Table S3). Wollega sesame variety produced a significantly greater seed yield (kg/ha) under WW condition followed by humera-1 and gondar-1 varieties. Correspondingly, sesame variety of gondar-1 had better seed yield than others followed by humera-1 under WS condition. However, the variation among the varieties in seed yield (kg/ha) under WS condition was insignificant. Wollega sesame variety had higher harvest index under WW

**Table 4  Two ways ANOVA tests on the effects of variety, water levels and their interaction on yield and yield components of sesame varieties.**

| Source | Dependent variable | df | Mean square | F | Sig. | Partial eta squared |
|---|---|---|---|---|---|---|
| Variety | Biological yield | 3 | 18.5 | 14.0 | .000 | 0.4 |
| | Harvest index | 3 | 67.7 | 0.236 | .871 | 0.0 |
| | Number of pods per plant | 3 | 12.9 | 5.5 | .002 | 0.2 |
| | Shoot biomass | 3 | 21.8 | 16.8 | .000 | 0.4 |
| | Root biomass | 3 | 0.4 | 12.3 | .000 | 0.4 |
| | Ys | 3 | 217,777.8 | 4.8 | 0.005 | 0.2 |
| | Yp | 3 | 7,161,666.7 | 28.8 | 0.000 | 0.6 |
| Water levels | Biological yield | 1 | 201.1 | 152.6 | .000 | 0.7 |
| | Harvest index | 1 | 1.1 | 0.0 | .952 | 0.0 |
| | Number of pods per plant | 1 | 296.1 | 124.7 | .000 | 0.7 |
| | Shoot biomass | 1 | 149.5 | 115.5 | .000 | 0.6 |
| | Root biomass | 1 | 3.2 | 109.8 | .000 | 0.6 |
| | Ys | 1 | 49,335,555.6 | 1,078.7 | 0.000 | 0.9 |
| | Yp | 1 | 25,7645,000.0 | 1,034.5 | 0.000 | 0.9 |
| Variety * Water levels | Biological yield | 3 | 5.4 | 4.1 | .010 | 0.2 |
| | Harvest index | 3 | 1,439.6 | 5.0 | .003 | 0.2 |
| | Number of pods per plant | 3 | 2.9 | 1.2 | .314 | 0.1 |
| | Shoot biomass | 3 | 4.317 | 3.3 | .025 | 0.1 |
| | Root biomass | 3 | 0.1 | 4.3 | .007 | 0.2 |
| | Ys | 3 | 217,777.8 | 4.8 | 0.005 | 0.2 |
| | Yp | 3 | 7,161,666.7 | 28.8 | 0.000 | 0.6 |

**Notes.**

$Yp$, Yield under well-watered condition; $Ys$, Yield under water-stressed condition.

condition attributed to the highest seed yield (kg/ha), while a relatively higher harvest index was recorded under WS condition in abasena and gondar-1 varieties, which might be due to the lower biological yield caused by the WS condition. Plants of abasena performed the least in most parameters under both water levels (Table 3, Table S3).

When variety, water levels and their interactive effects were analyzed, they produced a statistically significant difference in biological yield, number of pods per plant, Ys, Yp, shoot and root biomass at $p < 0.01$ (Table 4). Harvest index was not significantly affected by variety and water levels. Similarly, the interactive effect (variety * water levels) had no remarkable effect on number of pods per plant. The partial eta squared value revealed that water levels had greater effect on Ys, Yp, biological yield, number of pods per plant, shoot and root biomass followed by variety (Table 4).

## Drought tolerance indices

Analysis of variance of Ys, Yp and drought tolerance indices showed a highly significant difference among sesame varieties under the different water levels, indicating the presence of high genetic variability. Drought tolerance indices were calculated on the basis of seed yield (kg/ha) under WW (Yp) and WS (Ys) conditions. The mean seed yield under WW conditions ranged from 2,144.44 to 5,177.78 kg/ha, while 1,344.44 to 1,811.11 kg/ha was

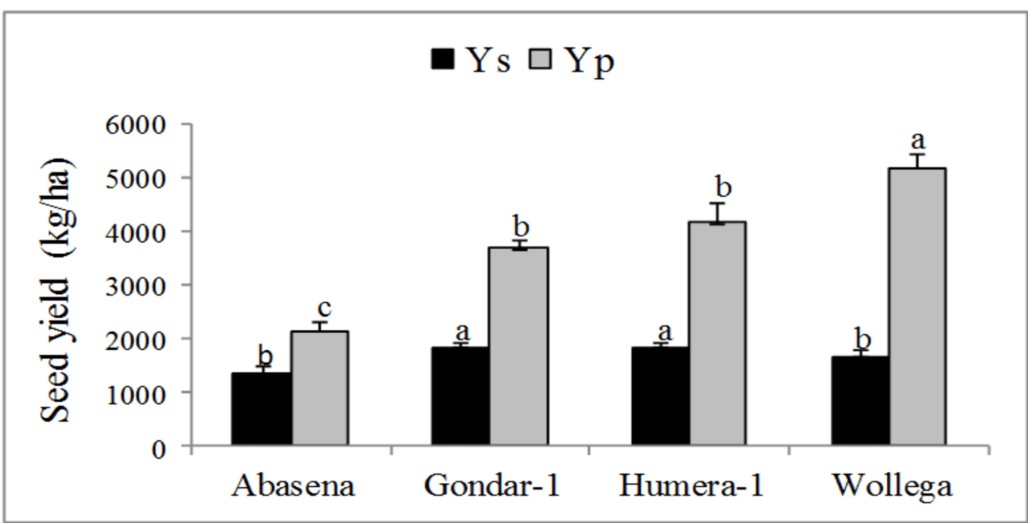

**Figure 2** **Mean seed yield of sesame varieties grown under water-stressed (Ys) and well-watered conditions (Yp).** Mean values followed by different letters are statistically significant at $p < 0.05$. Mean $\pm$ SE ($n = 9$).

obtained under WS condition. Wollega sesame variety had the highest and the second lowest mean seed yield (kg/ha) under WW and WS conditions, respectively (Fig. 2). Gondar-1 and humera-1 varieties achieved the highest seed yield (kg/ha) under WS condition. In both water levels, plants of abasena performed the least and the variation in yield (kg/ha) between WW and WS conditions was found to be significantly lower than the differences seen in other sesame varieties (Fig. 2).

The sesame varieties in this study showed a significant difference in drought tolerance indices (Table 5). Wollega sesame variety had significantly higher MP, GMP and TOL than others followed by humera-1 variety. According to these indices, wollega and humera-1 varieties were found to be more desirable and comparatively drought tolerant. Unlike this, wollega variety had lower STI than gondar-1 and humera-1 varieties (Table 5). The mean STI values demonstrated that gondar-1 and humera-1 had better drought tolerance characteristics. In the present study, abasena sesame variety had the lowest TOL, SSI and the highest YI and YSI of all the study varieties (Table 5). The difference between Ys and Yp values in abasena variety was the lowest indicating the lower sensitivity of the variety to drought stress. Varieties with lower TOL, SSI, and higher YI, YSI values are more stable and tolerant, respectively under drought conditions. Based on these indices thus, abasena could be identified as stable and tolerant variety. Plants of wollega variety demonstrated a significantly higher percent reduction than abasena that showed the lowest reduction. The studied varieties revealed a significantly different stress intensity of which the weakest was seen in abasena variety. Similarly, plants of gondar-1 and humera-1 had higher YSSI and YPSI values compared to others (Table 5).

**Table 5** Drought tolerance index values of sesame varieties grown under well-watered (WW) and water-stressed (WS) conditions.

| Drought indices | Sesame varieties | | | |
| --- | --- | --- | --- | --- |
| | Abasena | Gondar-1 | Humera-1 | Wollega |
| MP | 1744.44 ± 67.41[c] | 2738.89 ± 78.07[b] | 2977.78 ± 181.07[ab] | 3416.67 ± 136.49[b] |
| STI | 1310.91 ± 96.59[b] | 1810.89 ± 100.28[b] | 1809.94 ± 180.26[b] | 1659.86 ± 152.93[b] |
| GMP | 1667.12 ± 63.07[c] | 2568.40 ± 73.58[b] | 2709.39 ± 141.57[ab] | 2906.88 ± 134.41[b] |
| TOL | 800.00 ± 232.74[c] | 1855.56 ± 157.33[b] | 2333.33 ± 365.53[b] | 3522.22 ± 251.68[b] |
| YI | 0.63 ± 0.05[b] | 0.49 ± 0.02[b] | 0.44 ± 0.02[bc] | 0.32 ± 0.02[c] |
| YSI | 0.66 ± 0.08[b] | 0.50 ± 0.03[ab] | 0.48 ± 0.07[ab] | 0.32 ± 0.03[b] |
| % reduction | 33.84 ± 7.95[b] | 50.00 ± 2.94[ab] | 52.07 ± 6.95[ab] | 67.52 ± 2.93[b] |
| SSI | 0.91 ± 0.21[b] | 0.99 ± 0.06[b] | 0.93 ± 0.12[b] | 0.99 ± 0.04[b] |
| SI | 0.37 ± 0.00[d] | 0.51 ± 0.00[c] | 0.56 ± 0.00[b] | 0.68 ± 0.00[b] |
| YSSI | 655.91 ± 46.25[b] | 905.94 ± 50.14[b] | 905.43 ± 90.17[b] | 830.43 ± 76.45[b] |
| YPSI | 1127.22 ± 142.67[ab] | 1346.62 ± 96.11[b] | 1226.73 ± 98.94[ab] | 776.66 ± 152.20[b] |

**Notes.**

Mean values in a row that share different letters are significantly different at $p < 0.05$. (Mean ± SE and $n = 9$).

MP, Mean productivity; STI, Stress tolerance index; GMP, Geometric mean productivity; TOL, Drought tolerance; YI, Yield index; YSI, Yield stability index; SSI, Stress susceptibility index; SI, Stress intensity; YSSI, Yield stress score index; YPSI, Yield potential score index.

## Correlation

A strong association between Yp, Ys and drought tolerance indices is an important indicator to select the most desirable drought tolerance indices. Ys had positive and significant correlation with Yp. Moreover, Ys revealed a significantly positive correlation with MP, STI, GMP, YSSI and YPSI at $p < 0.01$ (Table 6). Similarly, Yp had significantly positive association with MP, STI, GMP, TOL, percent reduction, SI and YSSI at $p < 0.01$ probability level (Table 6). This showed that MP, STI, GMP and YSSI had significantly higher correlation with yield under WW and WS conditions so that these indices can be used as criteria to select varieties for drought tolerance. According to MP and GMP values, wollega and humera-1 were found to be the most desirable and drought tolerant varieties, respectively, while gondar-1 and humera-1 had higher STI and YSSI indicating higher yield performance under drought conditions. Furthermore, TOL had positive and significant correlation with MP, GMP, percent reduction, SSI and SI at $p < 0.01$. Its correlation with YI, YSI and YPSI was negative and significant (Table 6).

## Principal component analysis

The minimum number of principal components that accounted for most of the variations in drought tolerance indices was determined based on eigen value. Principal components with eigen values greater than 1 were selected. Accordingly, two principal components fulfilled the acceptable level/variance of the dataset. The first and second principal components (PC1) and (PC2) explained 86.2% and 12.7% of the variation, respectively. The PC1 and PC2 together covered 98.9% of the variation in the dataset (Table 7). The results of the analysis showed that PC1 had large positive association with Yp, TOL, MP and GMP. Others such as YI, YSI and YPSI had negative correlation with PC1. PC2 had large positive association with YPSI, Ys, STI, GMP, MP and YSSI, while negatively correlated with TOL

Table 6 Correlation coefficient between drought tolerance indices with seed yield under well-watered (WW) and water-stressed (WS) growing conditions.

| | Ys | Yp | MP | STI | GMP | TOL | YI | YSI | % reduction | SSI | SI | YSSI | YPSI |
|---|---|---|---|---|---|---|---|---|---|---|---|---|---|
| Ys | 1 | 0.28 | 0.50** | 0.74** | 0.68** | 0.01 | 0.24 | 0.25 | −0.25 | −0.52** | 0.34* | 0.74** | 0.80** |
| Yp | 0.28 | 1 | 0.97** | 0.59** | 0.89** | 0.96** | −0.70** | −0.80** | 0.80** | 0.38* | 0.85** | 0.59** | −0.29 |
| MP | 0.50** | 0.97** | 1 | 0.72** | 0.97** | 0.88** | −0.57** | −0.66** | 0.66** | 0.22 | 0.85** | 0.72** | −0.06 |
| STI | 0.74** | 0.59** | 0.72** | 1 | 0.81** | 0.41* | 0.09 | −0.24 | 0.24 | 0.06 | 0.29 | 1.00** | 0.53** |
| GMP | 0.68** | 0.89** | 0.97** | 0.81** | 1 | 0.74** | −0.43** | −0.52** | 0.52** | 0.10 | 0.79** | 0.81** | 0.16 |
| TOL | 0.01 | 0.96** | 0.88** | 0.41* | 0.74** | 1 | −0.79** | −0.90** | 0.90** | 0.54** | 0.79** | 0.41* | −0.52** |
| YI | 0.24 | −0.70** | −0.57** | 0.09 | −0.43** | −0.79** | 1 | 0.84** | −0.84** | −0.56** | −0.77** | 0.09 | 0.74** |
| YSI | 0.247 | −0.80** | −0.66** | −0.24 | −0.52** | −0.90** | 0.84** | 1 | −1.00** | −0.83** | −0.60** | −0.24 | 0.65** |
| %reduction | −0.25 | 0.80** | 0.66** | 0.24 | 0.52** | 0.90** | −0.84** | −1.00** | 1 | 0.83** | 0.59** | 0.24 | −0.65** |
| SSI | −0.52** | 0.38* | 0.22 | 0.06 | 0.10 | 0.54** | −0.56** | −0.83** | 0.83** | 1 | 0.10 | 0.06 | −0.59** |
| SI | 0.34* | 0.85** | 0.85** | 0.29 | 0.79** | 0.79** | −0.77** | −0.60** | 0.59** | 0.10 | 1 | 0.29 | −0.29 |
| YSSI | 0.74** | 0.59** | 0.75** | 1.00** | 0.81** | 0.41* | 0.09 | −0.24 | 0.24 | 0.10 | 0.29 | 1 | 0.53** |
| YPSI | 0.80** | −0.29 | −0.06 | 0.53** | 0.16 | −0.52** | 0.74** | 0.65** | −0.65** | −0.59** | −0.29 | 0.53** | 1 |

Notes.

** and * indicate significant differences at $p < 0.01$ and $p < 0.05$ probability levels, respectively.

MP, Mean productivity; STI, Stress tolerance index; GMP, Geometric mean productivity; TOL, Drought tolerance; YI, Yield index; YSI, Yield stability index; SSI, Stress susceptibility index; SI, Stress intensity; YSSI, Yield stress score index; YPSI, Yield potential score index.

Table 7 Eigen analysis of the correlation matrix and weight of each parameter.

| | PC1 | PC2 | PC3 | PC4 | PC5 | PC6 | PC7 | PC8 | PC9 | PC10 | PC11 | PC12 | PC13 |
|---|---|---|---|---|---|---|---|---|---|---|---|---|---|
| Eigen value | 2.88 | 1.68 | 0.76 | 0.50 | 0.20 | 0.09 | 0.06 | 0.02 | 0.00 | 0.00 | 0.00 | 0.00 | 0.00 |
| Proportion of variance | 0.862 | 0.127 | 0.011 | 0.00 | 0.00 | 0.00 | 0.00 | 0.00 | 0.00 | 0.00 | 0.00 | 0.00 | 0.00 |
| Cumulative proportion | 0.862 | 0.989 | 0.999 | 1.00 | 1.00 | 1.00 | 1.00 | 1.00 | 1.00 | 1.00 | 1.00 | 1.00 | 1.00 |
| | Cum.(%) | Ys | Yp | MP | STI | GMP | TOL | YI | YSI | % reduction | SSI | SI | YSSI | YPSI |
| Principal component 1 | 86.24 | 0.05 | 0.65 | 0.35 | 0.13 | 0.25 | 0.60 | −0.00 | −0.00 | 0.01 | 0.00 | −0.00 | 0.07 | −0.07 |
| Principal component 2 | 98.86 | 0.42 | 0.03 | 0.23 | 0.42 | 0.34 | −0.40 | 0.00 | 0.00 | −0.01 | −0.00 | −0.00 | 0.21 | 0.52 |

Notes.

MP, Mean productivity; STI, Stress tolerance index; GMP, Geometric mean productivity; TOL, Drought tolerance; YI, Yield index; YSI, Yield stability index; SSI, Stress susceptibility index; SI, Stress intensity; YSSI, Yield stress score index; YPSI, Yield potential score index.

(Table 7). The large positive and negative values of drought tolerance indices indicate the strong effect of each index on each principal component.

The biplot analysis showed that Yp, TOL, GMP, MP, Ys and YSSI orient in similar direction and had tight angles indicating a positive correlation and strong effect on PC1. Similarly, YPSI, Ys, STI and GMP contributed higher proportion to PC2 (Fig. 3). The biplot analysis clustered the study varieties roughly into two groups as low yielding (abasena) and relatively above average performing varieties (humera-1, gondar-1 and wollega) under WW and WS conditions (Fig. 3). This was corroborated with the values of STI, MP and GMP where the varieties in the second group had higher average yield values under WW and WS conditions (Table 7 and Fig. 3).

## Ranking of sesame varieties

Rank sum was calculated due to the fact that identification of genotypes for drought tolerance using individual indices is difficult. Different indices identified different sesame

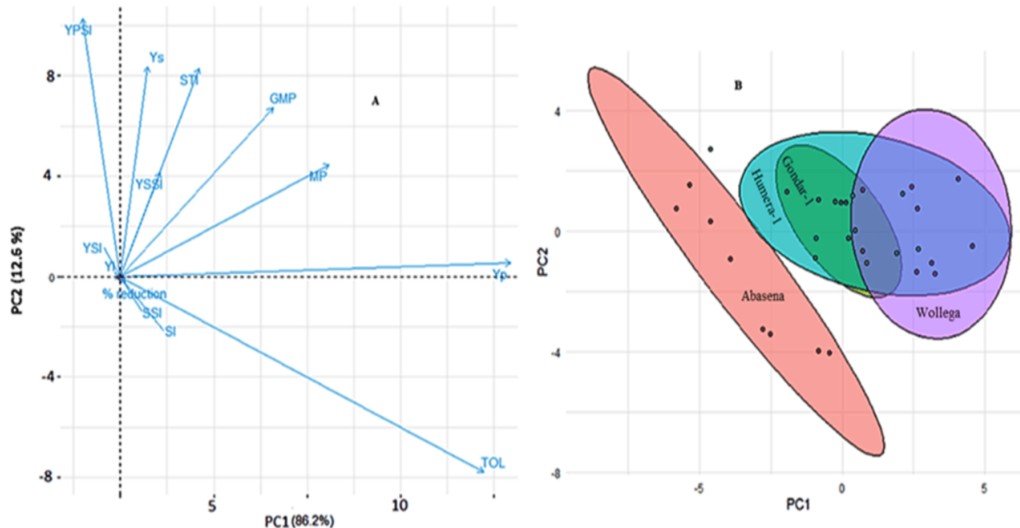

**Figure 3** Principal component analysis of drought tolerance indices (A) and biplot analysis of sesame varieties grown under well-watered (Yp) and water-stressed (Ys) conditions.

**Table 8 Rank mean, standard deviation of rank and rank sum derived from all drought tolerance indices.**

| Varieties | Rank mean | Standard deviation of rank (SDR) | Rank sum (RS) |
|-----------|-----------|----------------------------------|---------------|
| Abasena | 3.46 | 1.13 | 4.59 |
| Gondar-1 | 2.15 | 0.90 | 3.05 |
| Humera-1 | 2.08 | 0.64 | 2.72 |
| Wollega | 2.15 | 1.35 | 3.5 |

varieties for drought tolerance. Rank sum for all indices was used as an indicator to select the best varieties. Lower and higher rank sum values indicate high drought tolerant and susceptible genotypes, respectively. Accordingly, humera-1 sesame variety had the lowest rank sum value followed by gondar-1 variety. These varieties represented the best varieties with the highest performance under WW and WS conditions (Table 8). The rank sum result identified abasena sesame variety as the most susceptible and low yielding variety (Table 8).

## DISCUSSION

The results of the present study showed that the different agromorphological traits significantly varied among varieties due to the water levels. The results revealed that water stress had a remarkable influence on growth parameters specifically plant height, leaf number and number of branches per plant unlike physiological processes. The growth of branches per plant was completely suppressed under WS condition in all varieties. Humera-1 and wollega varieties showed the highest reduction in growth parameters. Gondar-1 outperformed all varieties in agromorphological traits, while abasena showed

the lowest. This is in line with the findings of *Hassen (2022a)*, *Hassen (2022b)*, *Mawcha et al. (2020)*, *Mekonnen & Sintayehu (2020)* and *Gebremichael & Parzies (2011)*. *Hassen (2022a)* and *Hassen (2022b)* has reported high heritability of plant height and harvest index of 100 sesame genotypes at Amibara, Ethiopia. Furthermore, nearly similar results of leaf length and number of branches per plant have been found in gondar-1, humera-1, wollega and argene sesame varieties (*Mawcha et al., 2020*). However, these and other varieties have shown a significantly higher plant height and number of leaves per plant (*Mekonnen & Sintayehu, 2020*; *Mawcha et al., 2020*). This deviation might be caused by the difference in the type of irrigation system, time of drought imposition, agroecology of the experimental sites that vary in soil fertility and other environmental factors. Furthermore, the findings have indicated that the agromorphological performance of sesame varieties varies across seasons (*Hailu et al., 2018*; *Hassen, 2022a*; *Hassen, 2022b*; *Baraki et al., 2020*).

The data in the present study demonstrated a significant reduction in yield and yield related traits under water stress. The highest reduction was recorded from wollega variety followed by humera-1. Gondar-1 had greater number of pods per plant and yield under WS condition followed by humera-1. Wollega variety had greater seed yield under WW condition. Almost similar, lower and higher yield performances of sesame varieties have been reported so far. According to *Mawcha et al. (2020)* gondar-1, humera-1 and wollega sesame varieties have produced seed yield of 3,432.09, 2,194.44 and 2,377.78 kg/ha, respectively, at Humera under supplementary irrigation; a common sesame production area, which is almost similar to the present study (Fig. 2). On the other hand, gondar-1 and humera-1 varieties produced significantly higher yield in the present study (Fig. 2) than the same varieties at Dansha grown under normal moisture conditions (588 kg/ha and 542 kg/ha), respectively. This higher yield deviation might be attributed to the high prevalence of bacterial blight disease, as the area is a hotspot for the disease (*Golla, Kebede & Kindeya, 2020*). A higher seed yield of abasena variety was obtained under WW condition (Fig. 2) in the present study compared to *Mekonnen & Sintayehu (2020)*. The authors have reported 1,840 kg/ha and 670 kg/ha seed yield under uniform optimum irrigation and 50% uniform deficit irrigation, respectively at Metema; another hub of sesame production. The yield of the variety under WS condition (Fig. 2) has been lower than 75% water deficit imposed at the vegetative development stage, 1,785 kg/ha (*Mekonnen & Sintayehu, 2020*). Comparatively, close yield performance as to the present study has been found in adi sesame variety treated with 100, 75 and 50% of the evapotranspiration of the crop applied through convention furrow irrigation method in 2015 at Werer research center (*Hailu et al., 2018*). On the other hand, the study varieties produced significantly greater seed yield than several other varieties and accessions such as serkamo white, adi, acc-00048, acc-00016, acc-00025, acc-00049 and others that have been evaluated at Werer (Afar Region), Bonta (Afar Region) and Miesso (Oromia) under normal growth conditions (*Hassen, 2022a* and *Hassen, 2022b*). This result was corroborated by *Baraki et al. (2020)* and found to be due to environmental and genetic variation among the sesame varieties. The authors have pointed out that about 42.62, 6.22 and 25.09% of sesame agronomic performance were determined by the environment, genotype and their interactive effect, respectively (*Baraki et al., 2020*).

Selection of crop varieties for drought tolerance using drought tolerance indices based on yield under normal and stressed conditions is one of the tools widely used in agriculture. In this study, wollega sesame variety had the highest MP, GMP and TOL values, which were consistent with the findings of *Baghery et al. (2022)* and *Farshadfar, Jamshidi & Aghaee (2012)*. The highest TOL value indicates the high sensitivity of the variety to water stress. This was evident from the highest percent reduction under water-stressed condition (Table 2). On the other hand, gondar-1 and humera-1 varieties had higher STI values showing better performance than other varieties. Several studies such as *Pireivatlou, Masjedlou & Aliyev (2010)*, *Farshadfar, Jamshidi & Aghaee (2012)*, *Zare (2012)*, *Noorifarjam, Farshadfar & Saeidi (2013)* and *Baghery et al. (2022)* have identified STI as a reliable index in wheat, wheat, barley, wheat and sesame, respectively, for selecting varieties with high drought tolerance and yield under normal and water-stressed conditions. This might be due to the fact that STI considers potential yield under normal condition, yield under stressful environments and stress intensity (*Baghery et al., 2022*). The abasena had the lowest TOL and SSI and the highest YI and YSI compared to the other varieties. The TOL and SSI values for the variety showed the lower yield potential under non-stress condition and higher yield under stressed condition, evidently proved by the lowest percent reduction under the two contrasting conditions (Table 2). The variety was also characterized by a closer Yp and Ys, implying the lower sensitivity of the variety to water-stressed condition, which subsequently resulted in smaller SSI. This result was consistent with the findings of *Zare (2012)* in barley, and *Baghery et al. (2022)* in sesame varieties.

The best drought tolerance index is the one that has discernable association with yield under well-watered and water-stressed conditions. In the present study, MP, GMP and STI had a significantly higher positive correlation with Yp and Ys. This shows that these indices are effective in identifying varieties under different water stress conditions as supported by the findings of *Siahsar, Ganjali & Allahdoo (2010)* in lentil. Moreover, Yp had significantly higher positive correlation with TOL, SI and YSSI and, Ys with YSSI and YPSI. These findings are in line with the reports of *Zare (2012)*, *Farshadfar, Jamshidi & Aghaee (2012)*, *Noorifarjam, Farshadfar & Saeidi (2013)*, *Baghery et al. (2022)* and *Sun et al. (2023)*. Drought tolerance indices correlated with both Yp and Ys have been found suitable for the selection of varieties for water stress (*Baghery et al., 2022*), indicating increase in yield under WW and WS conditions (*Farshadfar, Jamshidi & Aghaee, 2012*). The result of the PCA also revealed that only PC1 and PC2 with eigen value greater than 1 explained 98.9% of the variation, which was congruent with the reports of *Zare (2012)* in barley and *Baghery et al. (2022)* in sesame. According to the analysis, Yp, TOL, MP and GMP had a relatively strong effect on PC1 indicating the high yield potential and PC2 strongly associated to YPSI, Ys, STI and GMP, which are predictors of drought tolerance. Similar results have been reported in barley (*Zare, 2012*), sesame (*Baghery et al., 2022*) and cotton (*Sun et al., 2023*). Furthermore, the biplot analysis identified two categories of sesame varieties; low yielding (abasena) and above average performing varieties such gondar-1, humera-1 and wollega (Fig. 3). This corresponds with the categorization of *Sofi et al. (2018)*, where abasena roughly belongs to group D (poor yield performance under WW and WS conditions), wollega to group B (good performance under WW, not under WS conditions)

and gondar-1 and humera-1 varieties to group A (relatively higher performance in both WW and WS conditions. This might be related to the agroecology where the varieties are released that was characterized by high temperature and transpiration leading to water stress in the plant.

Since identifying the most drought tolerant variety using the indices was difficult, the rank sum and standard deviation of ranks of all indices were calculated. Accordingly, humera-1 followed by gondar-1 were identified as drought tolerant and high yielding, while abasena as the most susceptible and low yielding variety. Similar findings for other species and varieties have been reported by *Noorifarjam, Farshadfar & Saeidi (2013)* and *Anter & Ashraf (2018)*.

## CONCLUSION

The present study aimed to evaluate the agromorphological and yield performance of sesame varieties and to identify drought tolerant varieties using drought tolerance indices. Our findings showed a significant variation in agromorphological, seed yield and drought tolerance indices due to water levels, variety and their interaction. However, the effect of water levels was stronger than others. To support this result from the study, a significant reduction was observed in growth and yield parameters under water-stressed condition. The highest reduction was recorded in wollega. Abasena performed the lowest under both water levels. Humera-1 and gondar-1 varieties performed better both under water levels. They also had higher stress tolerance index (STI), yield stress score index (YSSI), yield potential score index (YPSI), geometric mean productivity (GMP) and mean productivity (MP) implying better tolerance to water stress. In general, humera-1 followed by gondar-1 are identified as high yielding and drought tolerant varieties based on drought tolerance indices, biplot analysis and rank sum of all drought tolerance indices. Therefore, STI, YSSI, YPSI, GMP and MP could be used for further sesame screening studies in areas with similar agroecology. Generally, the wider production of humera-1 and gondar-1 sesame varieties is recommended in and around the study areas.

### Funding

This work was supported by Bahir Dar University, College of Science, Office of Postgraduate, Research and Community service. The funders had no role in study design, data collection and analysis, decision to publish, or preparation of the manuscript.

### Grant Disclosures

The following grant information was disclosed by the authors:
Bahir Dar University, College of Science, Office of Postgraduate, Research and Community service.

### Competing Interests

The authors declare there are no competing interests.

## Author Contributions

- Getahun Yemata conceived and designed the experiments, analyzed the data, prepared figures and/or tables, authored or reviewed drafts of the article, and approved the final draft.
- Tewachew Bekele performed the experiments, authored or reviewed drafts of the article, and approved the final draft.

## Data Availability

The raw data are available in the Supplemental Files.

## Supplemental Information

Supplemental information for this article can be found online at http://dx.doi.org/10.7717/peerj.16840#supplemental-information.

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
