# Peer review of "Evaluation of sesame (Sesamum indicum L.) varieties for drought tolerance using agromorphological traits and drought tolerance indices"

_PeerJ, doi:10.7717/peerj.16840_

## Round 0.1 · original submission · Major Revisions

Dear Dr. Yemata,

Please address the reviewers' comments.

Reviewer 3 has requested that you cite specific references. You may add them if you believe they are especially relevant. However, I do not expect you to include these citations, and if you do not include them, this will not influence my decision.

·

Basic reporting

The study “Evaluation of sesame (Sesamum indicum L.) varieties for drought tolerance using agro-morphological traits and drought tolerance indices” by Yemata and Bekele is of scientific importance and provides insight into the responses of four sesame varieties to drought stress conditions. The authors have made efforts to find out the effect of drought stress on various agro-morphological, yield, and yield-contributing traits using various tolerance indexes and correlation analysis. Furthermore, a PCA was also carried out. The hypothesis and prime objective of the study are not clear and need to be included in the introduction section, as in the present manuscript, both hypothesis and objectives are indistinguishable. In addition, I urge authors to take care of punctuation.

Experimental design

The used methodology is appropriate; however, there is a lack of information regarding how the stress conditions were imposed (specify either crop stage), for how many days the crop remains under stress conditions, and what the level of stress is at critical crop growth stages such as flowering, seed formation, filling, and maturation. All these questions need to be answered.
Why did the authors not include test weight as a parameter? Test weight is an important parameter to find out the effect of drought stress on seed filling and assimilate partitioning.
Authors should also clearly mention at what stages the different parameters were recorded, e.g., at what stage and how many times during crop growth was RWC recorded? Which leaf was selected for the measurement of RWC?

Validity of the findings

Results
The results section is written in an appropriate manner; however, it would be better if the authors mentioned the range and mean of each trait for better understanding and comparison (WW and WS).
Discussion
The discussion part is well supported with the latest and relevant references. But still, it could be written in a more interesting way.

Additional comments

For more comments, please refer to the edited PDF file.

·

Basic reporting

All in all, the article looks in better shape and is rather well written. The main aspects of this publication the idea, theme relevance, novelty, importance of research object, and impact of results/findings all look solid. The findings of this work contribute well to the relevant field. The article is written in clear and good English. The literature is well-cited wherever required. I have only minor remarks that can be easily addressed.

Experimental design

The experimental design is well-defined. The material and methods should only contain the procedure of how the experiment was done. Line 158 is the result of soil analysis and should be included in the results not in material and methods.

Validity of the findings

Line 14 “The highest yield reduction was recorded in wollega, while the lowest was in abasena” should be made clearer like in which condition? WW or WS.
Line 94 varietiesa is varieties.
Line 205 Fernandez should be italicized.
Line 258 Plants of Gondar-1 variety performed the highest in terms of the number of leaves, leaf width, and plant height under WW conditions at 30 and 60 days after water stress imposition. This line is not properly framed as it means the Gondar-1 variety performed well in terms of the number of leaves, leaf width, and plant height under WS conditions at 30 and 60 days after water stress imposition. Simply replace WW with WS.
Line 266 Are these percentages average of all the varieties? make it more clear.
Line 273 the correlation data should be included in the paper.
Line 282 grammatical error. Significant effect size than variety should be replaced with significant effect rather than variety.
Table 3 does not clear that at which water level data is presented.30 days, 60 days or average.

Additional comments

The figures are relevant, however, the quality of Figure 3 can be improved if possible.
Supplementary data should be mentioned in the main data otherwise, there is no point.
The in-text references should be of uniform format throughout.

Reviewer 3 ·

Basic reporting

Attached in the review report

Experimental design

Attached in the review report

Validity of the findings

Attached in the review report

Additional comments

The authors evaluated the sesame (Sesamum indicum L.) varieties for drought tolerance using
Agro-morphological traits and drought tolerance indices.

I do have several suggestions that need to be addressed by major revision. Below are some comments to be taken into consideration by the authors

General comments
-The manuscript presents interesting data with innovative findings that fall into the scope of the journal.
-The introduction is weakly written. Re-write it with an emphasis on abiotic stress by reviewing appropriate literature.
-The methodologies seem to be sound to interpret the data and draw a final and complete conclusion.
-The descriptions of mostly drought imposition in the field are not adequate to allow replication of the estimated methods properly.
- I found irrelevant citations in the manuscript. Check all citations for it relevancy thoroughly and omit as well as replace with relevant citations
-The English language needs to be improved throughout the whole manuscript.
-The whole discussion chapter should be improved well by rewriting it with adequate interpretation and explanation of the results chronologically, citing appropriate references.

Specific comments
Abstract
Line 16: Change “well watered” to “well-watered”. Change “water stressed” to “water-stressed”. Follow this style throughout the whole manuscript where it exists.
Line 10: Change “number of leaves” to “number of leaves per plant”. Follow this style throughout the whole manuscript where it exists including text, tables, and figures.
Line 23: Delete “in the study varieties.”
Line 24: Change “as compared to “compared to”. Follow this style throughout the whole manuscript where it exists.
Line 26: ---seed yield ha-1. The unit is incomplete. Change “seed yield ha-1” to “seed yield (kg ha-1)”.

Keywords
These should be changed. Avoid inclusion of variety names.

Introduction
The introduction is poorly written. They highlighted the crops, however, they escape abiotic stress effects, such as drought/salinity, etc. in the introduction. It must be improved. Re-write the introduction with an emphasis on abiotic stress by reviewing appropriate literature.
Line 55: Change “37°C” to “37 °C”. Follow this style throughout the whole manuscript where it exists including text, tables, and figures.
Line 58: Provide countries and world productions and the area of the crop.
Line 76: Add “Abiotic stress, such as drought/salinity affects the growth and productivity (doi.10.1002/jsfa.9423; doi: 10.3389/fpls.2022.992535), generate osmotic stress (doi.10.3389/fpls.2020.559876) and reactive oxygen species (ROS) (doi.10.1038/s41598-018-34944-0). It eventually creates oxidative impairment in plants (doi.10.1007/s12010-018-2784-5). Osmotic stress leads to reduction of photosynthetic activities (doi.10.1016/j.foodchem.2018.01.097), and nutrients imbalance including macro and micro-nutrients in plants (doi.10.1186/s12870-018-1484-1; doi.10.1371/journal.pone.0206388). It also triggers biochemical changes such as both enzymatic and non-enzymatic antioxidants, such as phenolic and flavonoids such as hydroxybenzoic acids (doi.10.3390/antiox11122434), flavanols (doi.10.3390/antiox12010173), flavanones (doi.10.3390/antiox11030578), hydroxycinnamic acids (doi.10.1038/s41598-018-30897-6), flavonols (doi.10.3389/fnut.2020.587257), flavones (doi.10.1038/s41598-020-71727-y), ascorbic acids (doi.10.1186/s12870-020-02780-y), and carotenoids (doi.10.3390/molecules27061821).
Line 94: Change “varietiesa” to “varieties”.

Materials and methods
-It is not clear how the author maintains field capacity and drought stress in the field condition. The description for pot soil. They should explain it well.
-There is a huge ambiguity in the name of varieties. In the abstract and introduction, the author mentioned the varieties as humera-1, gondar-1, wollega and abasena. However, in the materials and methods, they mentioned them as T-85, Kelafo-74, Mehado-80, and Abasena. Why the released varieties' names are changed by the people? The author should use the released names.
-Include weather data for the experimental location and period.
Line 100: Change “11°41ʹ51ʹʹN latitude and 36°56ʹ 35ʹʹ east longitude” to “11° 41ʹ 51ʹʹ north latitude and 36° 56ʹ 35ʹʹ east longitude”.
Line 103: Change “1500mm” to “1500 mm”.
Line 129-130: Change the lowercase letter x “39m2 (9.75m x 4m) consisting of 24 plots each with 1m2 area,” to the symbol of the cross “39 m2 (9.75 m × 4 m) consisting of 24 plots each with 1 m2 area,”.
Line 131: Change “0.5m” to “0.5 m”. There are numerous similar errors. Add space between the number and the unit. Follow this style throughout the whole manuscript where it exists including text, tables, and figures.
Line 150: Change “333ml” to “333 mL”.
Line 179: Change “24 hrs” to “24 hr”.
Line 180: Change “70°Cs” to “70 °C”.
Line 182: Change “(FW-DW) / (TW-DW)*100.” to “(FW-DW)/(TW-DW) × 100.”.
Line 198-199: Change to “Harvest index = (Seed yield/biological yield) × 100”.
Line 205-214: Change to “*” to the symbol of the cross “×”. Add space before and after the symbol “+” and delete the space before and after the symbol “/”
Line 223-224: delete the word “rank”.

Results
Put a subheading: “Growth and relative water content responses of sesame varieties” for Table 1 results and “analysis of variance for main and interaction effects of growth parameter” for Table 2 results
Line 273: WW conditions (Data not shown). According to journal criteria, the author must show the data. Otherwise, omit the sentence.
Table 1: Add space before and after the symbol “±”. Change “Number of branches” to “Number of branches per plant”. Change “Number of leaves” to “Number of leaves per plant”. Follow this style throughout the whole manuscript where it exists including text, tables, and figures.
Line 271-278: The correlation table number is 6. Why there is a correlation presentation after Table 1? Results must be presented chronologically following tables and figures. Write a section correlation separately after Table 5 and transfer this writing there.
Line 283-289: Repetition of previous results. Delete it.
Table 3: Add space before and after the symbols “±” and “=”.
Line 293: Change “Mean performance of sesame varieties” to “yield and its attributes of sesame varieties”.
Figure 2 and Table 4: Why there is a different abbreviation for well-watered and water stress. The units are not consistence with the unit of the text. Unified it throughout the text, tables, and figures.
Figure 3: Must be visible enough for readability.

Discussion
-The discussion chapter is very poor. The whole discussion chapter should be improved well by rewriting it with adequate interpretation and explanation of the results chronologically, citing appropriate references.
Line 403: Change “and genetic variability” to “and well-watered”.
Line 409-419: All the discussion is invalid as all the references are related to the heritability, variability, diversity, and stability of sesame. The author must include discussion and citations related to drought sesame or other crops. Follow and cite articles: doi.10.3390/su15021427; doi.10.3390/life12111816; doi: 10.3389/fpls.2022.992535.
It is too poor to read. I don’t go through the rest of the discussion, improve it substantially then I will see the revised version to make comments.

Conclusions
It should be rewritten with major findings, conclusive remarks, and recommendations for the study.

---

## Round 0.2 · Minor Revisions

Dear Dr Yemata,

The manuscript on 'Evaluation of ...... indices' needs to be thoroughly checked, Still I could see many corrections, that need to be done before its acceptance for publication. For example,

- most of the reference looks 1990's. There are many recent work/references on sesame drought/moisture stress tolerance published after 2020
- there are many spelling errors.
- in references: scientific names were in not italics.

I suggest authors look back into the again and correct it.

**Language Note:** The review process has identified that the English language must be improved. PeerJ can provide language editing services - please contact us at copyediting@peerj.com for pricing (be sure to provide your manuscript number and title). Alternatively, you should make your own arrangements to improve the language quality and provide details in your response letter. – PeerJ Staff

---

## Round 0.3 · Minor Revisions

Dear Authors,

Reviewer #1 made some comments in the manuscript attached, they seem to be reasonable and for the improvement of the manuscript.

with Regards

·

Basic reporting

The authors answered a few queries; however, most of the comments are unanswered.

Experimental design

Many comments are unanswered and needed author's response.

Validity of the findings

NA

Additional comments

Please refer the attachment for queries

·

Basic reporting

Following revisions, the article has been improved. The fundamental elements of this publication—concept, theme pertinence, originality, significance of research subject, and influence of results and findings—appear to be sound. This research contributes significantly to the pertinent field. Revision has resulted in an improvement to the English. The literature is appropriately referenced and current when applicable.

Experimental design

The experimental design is well defined. The material and methods should only contain the procedure how the experiment was done. Line 170 is the result of soil analysis and should be included in the results, not in the material and methods.

Validity of the findings

The quality of Figure 3 has been improved.
The in-text references are now in uniform format.

---

## Round 0.4 · accepted · Accept

Dear Dr. Yemata,
Thank you for your submission to PeerJ.
I am writing to inform you that your manuscript - Evaluation of sesame (Sesamum indicum L.) varieties for drought tolerance using agro-morphological traits and drought tolerance indices - has been Accepted for publication. Congratulations!